# MMAI Gym for Science: Training Liquid Foundation Models for Drug Discovery

**Maksim Kuznetsov[1], Zulfat Miftahutdinov[1], Rim Shayakhmetov[1], Mikolaj Mizera[1],
Roman Schutski[1], Bogdan Zagribelnyy[1], Ivan Ilin[1], Nikita Bondarev[1],
Thomas MacDougall[1], Mathieu Reymond[1], Mihir Bafna[2], Kaeli Kaymak-Loveless[2],
Eugene Babin[1], Maxim Malkov[1], Mathias Lechner[2], Ramin Hasani[2],
Alexander Amini[2], Vladimir Aladinskiy[1], Alex Aliper[1], Alex Zhavoronkov[1]**

[1] Insilico Medicine    [2] Liquid AI

## Abstract

General-purpose large language models (LLMs) that rely on in-context learning do not reliably deliver the scientific understanding and performance required for drug discovery tasks. Simply increasing model size or introducing reasoning tokens does not yield significant performance gains. To address this gap, we introduce the MMAI Gym for Science, a one-stop shop molecular data formats and modalities as well as task-specific reasoning, training, and benchmarking recipes designed to teach foundation models the "language of molecules" in order to solve practical drug discovery problems. We use MMAI Gym to train an efficient Liquid Foundation Model (LFM) for these applications, demonstrating that smaller, purpose-trained foundation models can outperform substantially larger general-purpose or specialist models on molecular benchmarks. Across essential drug discovery tasks – including molecular optimization, ADMET property prediction, retrosynthesis, drug–target activity prediction, and functional group reasoning – the resulting model achieves near specialist-level performance and, in the majority of settings, surpasses larger models, while remaining more efficient and broadly applicable in the domain.

## 1 Introduction

Large language models (LLMs) are increasingly being explored as tools for drug discovery (Taylor et al., 2022; Pei et al., 2023; Narayanan et al., 2025), either by fine-tuning general-purpose open-source models Wang et al. (2025) or by training domain-specific models for individual downstream tasks, such as property prediction, synthesis planning, and target prioritization (Edwards et al., 2022; Livne et al., 2024). Despite promising demonstrations, current LLM-based approaches typically lag behind specialist methods on core drug discovery benchmarks, and performance gains obtained via prompting or task-local fine-tuning often do not translate into robust, cross-task capability. As a result, it remains challenging to obtain a single "generalist" model that performs competitively in the diverse molecular reasoning workloads encountered in medicinal chemistry, biology, and early clinical development.

We propose a training recipe that uses supervised (SFT) and reinforcement learning (RFT) fine-tuning on domain-specific data to turn a general-purpose causal LLM into a drug-discovery generalist. We instantiate this approach in **MMAI Gym**, a structured training and evaluation environment that provides curated scientific reasoning traces across key modalities and tasks in drug discovery. Unlike approaches that rely primarily on generic reasoning reinforcement learning (RL) or fine-tuning on a single benchmark's training split, MMAI Gym emphasizes domain-faithful reasoning chains, task formats used by practitioners, and evaluation under distribution shift via held-out and out-of-distribution benchmarks. We use MMAI Gym to finetune the Liquid Foundation Model **LFM2-2.6B model** Amini et al. (2025), a highly efficient language model, into a powerful model for drug discovery, where it achieves competitive or state-of-the-art results across a diverse set of molecular prediction tasks. Our results show that a single SFT+RFT run substantially improves –

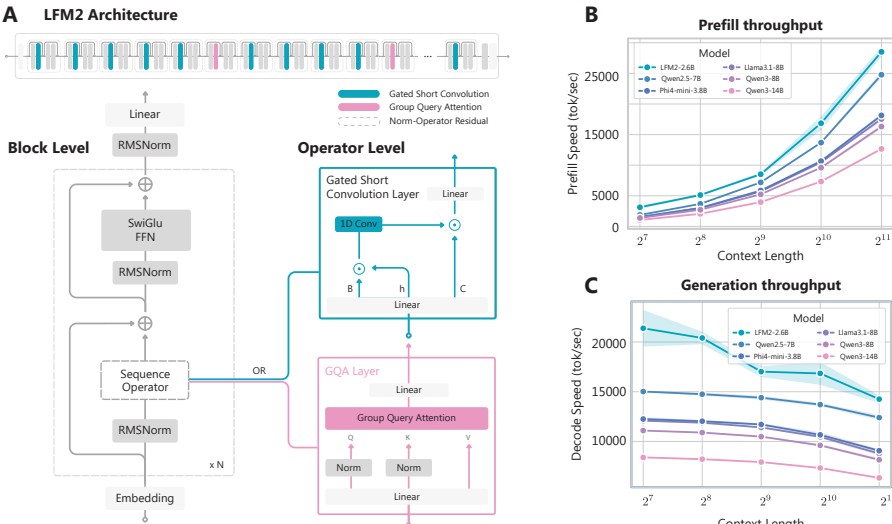

Figure 1: **LFM2 Architecture & Efficiency**. (a) LFM2 is a hybrid language model comprised mainly of gated short convolution sequence operators offering improved throughput and efficency (Amini et al., 2025). (b,c) LFM2 prefill processing speed and decode speed with increasing prefill.

in some cases by up to an order of magnitude on task metrics – LFM2's performance on selected chemistry-oriented benchmarks relevant to drug discovery and development. These results suggest that domain-specific reasoning supervision can be an effective path toward robust multi-task LLMs for scientific R&D.

## 2 LIQUID FOUNDATION MODEL

Full softmax attention (Vaswani et al., 2017) based language models incur substantial computational and memory costs, particularly for long-context and resource constrained deployment scenarios. These limitations motivate the need for efficient alternatives, including hybrid architectures in which the majority of layers use sub-quadratic sequence operators, such as linear attention or state space models (SSMs) (Dao & Gu, 2024; Yang et al., 2025; Katharopoulos et al., 2020), while a smaller number of softmax attention layers are interleaved to serve as global token mixers. This design has been shown to achieve a favorable Pareto tradeoff between model expressivity and efficiency (Team, 2025; Blakeman et al., 2025).

To this end, we utilize LFM2-2.6B (Amini et al., 2025), a 2.6B parameter autoregressive hybrid language model from the Liquid Foundation Model (LFM) family, specifically designed for long context inference under latency and memory constraints. LFM2-2.6B (Figure 1a) is composed of two sequence operator primitives: primarily *gated short convolutions* (ShortConv) for sequence featurization and a few *grouped-query softmax attention* (GQA) (Ainslie et al., 2023) for global sequence mixing.

Each LFM2 block is composed of two sequential `Norm-Operator-Residual` schemes: first, where the sequence operator is either a ShortConv or GQA, and second, where the operator is SwiGLU feed-forward layers (Shazeer, 2020) for the position-wise MLP readout. Each block utilizes a pre-Root Mean Squared Normalization (pre-RMSNorm) (Zhang & Sennrich, 2019; Jiang et al., 2023), which omits the centering step in LayerNorms for improved efficiency.

The gated short convolution operator, for a hidden sequence $h \in \mathbb{R}^{L \times d}$, computes the following,

$$(B, C, \tilde{h}) = hW_{\text{in}}, \qquad o = \left( C \odot \text{Conv}_k(B \odot \tilde{h}) \right) W_{\text{out}} \tag{1}$$

where $W_{\text{in}} \in \mathbb{R}^{d \times 3d}, W_{\text{out}} \in \mathbb{R}^{d \times d}$ are learned linear projections, $\text{Conv}_k$ denotes a depthwise 1D convolution with $k = 3$, and $\odot$ denotes the canonical Hadamard product. These local operators encode short range "motifs" in the sequence, which are subsequently integrated over long contexts by the small number of global attention operators. The global mixing layers use GQA (Ainslie et al., 2023), which partitions queries into groups that share keys and values, reducing overhead.

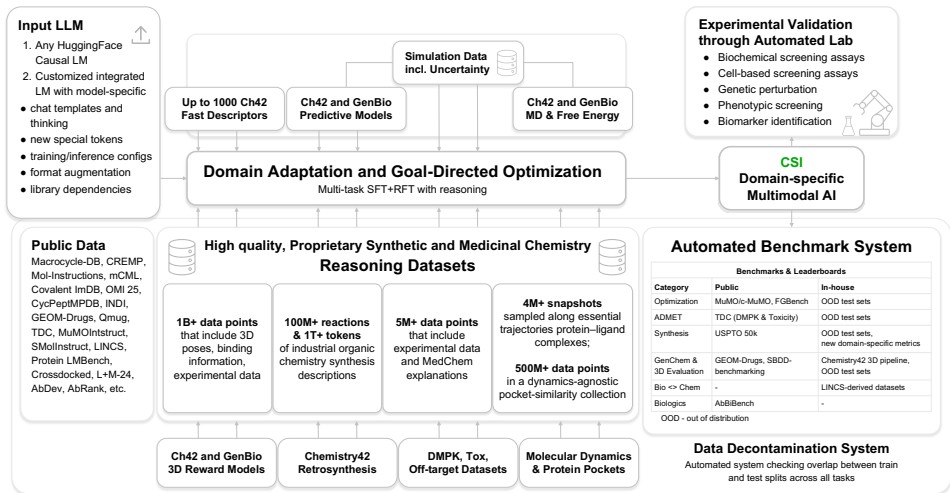

Figure 2: **MMAI Gym**: Integrated Data, Training, and Benchmark Suite

Empirically, these architectural design choices translate into substantial efficiency gains during both prefill and decode. Across a range of prefill context lengths, LFM2-2.6B consistently achieves higher throughput than the other smallest open-weight baseline models that were also finetuned using MMAI Gym, as well as Phi-4-mini (Abouelenin et al., 2025), another model optimized for compute, memory, and latency bound environments (Figure 1b,c). This trend reflects the contrast between the LFM hybrid design, with its linear-time ShortConv operators and limited GQA, and attention-dominant transformers that incur quadratic scaling at longer contexts.

## 3  MMAI Gym for Science

Deploying LLMs in chemistry, biology, and other domains requires the ability to handle very long context inputs efficiently (i.e., data from laboratory experiments), to perform robustly across a diversity of highly specialized downstream tasks (i.e., from predicting scalar toxicity values to designing experimental protocols), and to generalize to unseen scenarios and distribution shifts (i.e., new chemo-types and novel protein targets). We thus hypothesized that a domain-specific training approach would enable the efficient, long-context LFM2 architecture to achieve strong performance in scientific scenarios. To demonstrate this in the context of chemical drug discovery, we built and leveraged MMAI (Multi-Modal AI) Gym for Science, a suite of data, training, and evaluation environment designed to adapt and improve general-purpose and chemical domain-specific LLMs to molecular reasoning tasks primarily in drug discovery and development (Figure 2). While strong at generic language tasks, general-purpose frontier LLMs often fail on chemistry, biology, and clinical problems – e.g., ADME and toxicity endpoint prediction, chemical plausibility for retrosynthesis (Zagribelnyy et al., 2026), and clinically grounded decision tasks. Prompt engineering or naive fine-tuning typically yields limited robustness, especially under out-of-distribution shifts. MMAI Gym addresses these limitations by treating these problems as domain-specific reasoning tasks rather than pure text prediction, targeting the formats and causal chains used by medicinal chemists, biologists, and clinical researchers.

MMAI Gym implements a multi-stage curriculum combining (i) curated, domain-specific reasoning datasets spanning medicinal chemistry optimization chains, synthesis and reaction reasoning, drug metabolism, pharmacokinetics (DMPK), and toxicity modeling, and geometry-grounded tasks incorporating 3D structures; (ii) multi-task supervised and reinforcement fine-tuning; and (iii) automated data decontamination plus evaluation on public and internal out-of-distribution benchmarks. The intended outcome is a single adapted model that can jointly perform property prediction, design/optimization reasoning, synthesis planning, target/biology reasoning, and select clinical inference tasks with improved performance and reliability.

## 3.1 Datasets and Data Balancing

MMAI Gym comprises a suite of 200+ tasks spanning dozens of categories, the main high-level ones being 2D molecule tasks, 3D molecule tasks, 2D protein tasks, 3D protein tasks, drug–gene interaction tasks, and cross-domain tasks that operate jointly in multiple modalities. Each broad category covers a variety of task formulations, including predictive tasks with easily verifiable answers and molecular generation tasks where a large set of diverse answers is possible.

The molecular tasks are built from a range of datasets, including but not limited to TDC (Huang et al., 2021), MOSES (Polykovskiy et al., 2020), FGBench (Liu et al., 2025), MuMO-Instruct (Dey et al., 2025), CREED, URSA-expert-2026, USPTO-50K-test (Zagribelnyy et al., 2026), and MolTextNet (Zhu et al., 2025)—and the 3D molecular (spatial) tasks draw on GEOM (Axelrod & Gómez-Bombarelli, 2022), ZINC (Irwin & Shoichet, 2005), and QMUGS (Isert et al., 2022). Protein tasks leverage ProteinLMBench (Shen et al., 2024), the AlphaFold Protein Structure Database (Varadi et al., 2023), PepBDB (Wen et al., 2018), and the BC40 subset of PDB (Berman et al., 2000). Drug–gene and protein-protein interaction tasks are based on LINCS Subramanian et al. (2017) and StringDB Szklarczyk et al. (2024), while cross-domain tasks incorporate CrossDocked Francoeur et al. (2020), L+M-24 Edwards et al. (2024), MolInstructions (Fang et al., 2024), and SMolInstruct (Yu et al., 2024); across main categories, these datasets are complemented with high-quality in-house data.

To integrate the diverse datasets and modalities, we adopt a multi-stage balancing strategy. When constructing training batches, we first sample uniformly from the six top-level categories; within the selected category, we then sample a dataset-task uniformly and finally draw an example from that dataset. This procedure is designed to mitigate imbalances arising from differing dataset sizes across modalities and sources, ensuring that all tasks are represented uniformly within each category.

## 3.2 Tokenization and Augmentation of Chemical Entities

To enable LFM2 model to learn abstract molecular concepts as opposed to overfitting to a particular syntax of the molecular format, we use chemical format conversion as part of our data augmentation strategy. For each user input, we randomly convert SMILES Weininger (1988) to SELFIES Krenn et al. (2020) and vice versa; additionally, SMILES/SELFIES strings are randomly converted to IUPAC names when possible. This format augmentation helps the model to learn modality-independent molecular concepts and acts as linguistic bridge between valid chemistry and natural language. To further enhance chemical generalization, we also augment SMILES and SELFIES representations during training by applying non-canonical random traversal both for user input and model output.

Following prior chemical language models Livne et al. (2024); Pei et al. (2023), we augment LFM2's base natural language vocabulary with tokens specific to SMILES, SELFIES, and FASTA formats. This separation is intended to distinguish chemical symbols and ensure a consistent encoding of molecular strings. We additionally wrap each molecular sequence with format type tags. For example, an Acetic Acid is isolated as `<smiles>CC(=O)O</smiles>` and tokenized as :

```
[<smiles>,<sm_C>,<sm_C>,<sm_(>,<sm_=>,<sm_O>,<sm_)>,<sm_O>,</smiles>]
```

During training, we enforce this specialized format isolation and tokenization for model-generated outputs, while applying it to user inputs with probability 0.5 to better align the base model representations with the newly introduced molecular tokens. This allows the model to understand and convert user input that contains only standard text tokens, while having the power to represent the molecules for best performance.

To represent 3D molecular and protein structures, we follow the text-based representations introduced in nach0-pc Kuznetsov et al. (2025) and BindGPT Zholus et al. (2025). For small molecules, we first provide the molecular graph as a SMILES string, and then list the 3D coordinates of each atom in the same order as in the SMILES. For proteins, we lay out atomic coordinates residue-by-residue, annotating each residue with a dedicated amino-acid token (e.g., `<am_A>` for alanine, `<am_R>` for arginine, ..., `<am_V>` for valine) and each atom with an atom-name token (e.g., `<atom_name_CA>`, `<atom_name_CB>`, ..., `<atom_name_NH2>`). To optimize textual context usage, we omit hydrogens and reconstruct their positions post hoc using standard cheminformatics toolkits O'Boyle et al. (2011); Landrum et al. (2023).

### 3.3 Reinforcement Learning Rewards

All MMAI Gym tasks can be classified as 3 different kinds of tasks: (i) regression, (ii) classification, (iii) generation. To perform reinforcement learning post-training, each type incorporates a reward function, providing a score for the generated completion. MMAI Gym includes 2 generic reward functions applicable to all tasks. First, the *format* reward ensures the reasoning is properly enclosed in `<think>` tags. Formally, $r_{\text{format}} = -1 + \mathbb{1}_{n_{<\text{think}>}=1} + \mathbb{1}_{n_{<\backslash\text{think}>}=1}$, where $n_{<\text{think}>}$ counts the number of occurrences of `<think>`, and $\mathbb{1}$ is the indicator function. Second, the *thinking* reward encourages longer thinking/reasoning sequences. Among the large number of task-datasets present in MMAI Gym, some do not incorporate reasoning, resulting in sequences where the answer follows immediately after the question. This results in a behavior where the model sometimes output an answer immediately after a question, which hampers the sequences generated during reinforcement learning post-training. To mitigate this, we encourage the model to generate longer sequences, up to a saturation point of $5,000$ characters, i.e., $r_{\text{think}} = \min(1, -1 + |o|/2500)$.

In addition to these rewards, each task perceives its own task-specific reward. In each case, the reward function considers both an answer-formatting component and a correctness component.

**Classification.** All considered classification tasks are question-answering problems, where the model must choose between a set of possible answers. Given the ground-truth $g$ and answer $a$, $r_{\text{qa}} = \mathbb{1}_{||a|-|g||<3} + \mathbb{1}_{a=g} - 1$. The first part constraints the difference in length between the ground-truth and prediction to be less than 3 characters, ensuring that the model only outputs the provided choice (e.g., "True") without any additional justification.

**Regression.** For regression tasks, we define the reward as $r_{\text{reg}} = \mathbb{1}_{||a|-|g||<3} - \frac{|a-g|}{(\max(\mathcal{A})-\min(\mathcal{A}))}$, where $\mathcal{A}$ is the dataset of all the answers, and the second part corresponds to the normalized absolute value between the answer and the ground-truth. Analogously to $r_{\text{qa}}$, the first part encourages the predicted answer to have a similar format to the ground-truth (in this case, similar decimal precision).

**Molecular generation.** Finally, molecular generation tasks verify that the generated answer is a valid molecule as determined by RDKit Landrum et al. (2023), i.e. $r_{\text{gen\_only}} = 2 * \mathbb{1}_{\text{is\_valid(a)}} - 1$. Alternatively, if there exists a set of verified ground truth values $G$ for a given task, then presence in the ground-truth set supersedes the validity. The reward for these tasks is $r_{\text{gen\_gt}} = 2 * \mathbb{1}_{a \in G} - 1$.

## 4 Experiments

We trained the domain-adapted model, LFM2-2.6B-MMAI, on the MMAI Gym datasets and evaluated it on a variety of cheminformatic and drug discovery tasks, comparing to several baseline tiers. Training is performed in two stages: first using supervised fine-tuning (SFT), then applying reinforcement learning fine-tuning (RFT) in both (i) multi-domain multi-task (**MT**) and (ii) single-category or, simply, single-task (**ST**) settings. Across the experiments section, we consider single-category setting as "single-task" **ST** even if the category has multiple distinct tasks, e.g. ADMET Properties Prediction. Full training details are provided in Appendix A.

### 4.1 Benchmarking Protocol

For each benchmark task, every test example is evaluated multiple times. At each repetition, a prompt is assembled by randomly sampling from a pool of paraphrased instruction templates and the molecular input is independently augmented according to our augmentation and tokenization protocol, yielding a distribution of predictions over diverse prompt formulations, input augmentations, and stochastic decoding samples. The model generates a completion with a chain-of-thought reasoning trace enclosed in `<think>` tags, followed by an `<answer>` block. A parser extracts the prediction from the `<answer>` block, either a number for regression tasks, a class label for classification tasks, or a molecule for generation tasks. We additionally extract class probabilities from the first-token logprobs via a softmax over the tokens representing each class for metrics requiring class probabilities. The resulting distribution of predictions across repetitions is aggregated per example. We use median for continuous values, majority vote for categorical, or a task-specific aggregation method if the benchmark requires it. The aggregated value is a robust estimate that marginalizes over template phrasing, molecular representation, and model uncertainty.

| Model | BDP | | | BDQ | | | BPQ | | | DPQ | | | BDPQ | | |
|---|---|---|---|---|---|---|---|---|---|---|---|---|---|---|---|
| | SR | Sim | RI | SR | Sim | RI | SR | Sim | RI | SR | Sim | RI | SR | Sim | RI |
| *Foundational LLMs for Chemistry* | | | | | | | | | | | | | | | |
| ChemLLM | 0.2 | 0.17 | 1.2 | 1.0 | 0.55 | 0.82 | 4.8 | 0.29 | 0.96 | 0.6 | 0.28 | 0.42 | 0.0 | n/a | n/a |
| LlaSMol$_{Mistral}$ | 43.6 | 0.62 | 1.09 | 31.4 | **0.66** | 0.93 | 86 | 0.58 | 0.84 | 24 | **0.57** | 0.61 | 14 | 0.62 | 1.03 |
| *Task-specific non-LLMs* | | | | | | | | | | | | | | | |
| Prompt-MolOpt | 12.2 | 0.12 | **7.46** | 23.2 | 0.10 | 5.4 | 15.8 | 0.10 | 1.5 | 23.6 | 0.10 | **5.46** | 6.6 | 0.11 | **5.36** |
| *Task-specific LLMs* | | | | | | | | | | | | | | | |
| GeLLM$^3$O-3$_{Mistral}$ | 84.8 | 0.47 | 4.3 | 87.0 | 0.47 | 5.61 | 93.0 | 0.46 | 1.49 | 62.8 | 0.37 | 3.87 | – | – | – |
| GeLLM$^3$O-3$_{Llama}$ | **86.8** | 0.48 | 4.38 | 90.0 | 0.46 | **5.66** | 94.0 | 0.5 | 1.38 | 60.6 | 0.44 | 3.76 | – | – | – |
| GeLLM$^3$O-4$_{Mistral}$ | 71.6 | 0.49 | 3.27 | 57.4 | 0.55 | 2.56 | 90.2 | 0.46 | 1.41 | 54.0 | 0.44 | 3.02 | 30.0 | 0.48 | 3.44 |
| GeLLM$^3$O-4$_{Llama}$ | 53.6 | **0.63** | 1.94 | 48.6 | 0.59 | 1.29 | 93.4 | **0.59** | 1.12 | 39.6 | **0.57** | 1.32 | 28.0 | **0.66** | 1.02 |
| *Generalist LLMs* | | | | | | | | | | | | | | | |
| GeLLM$^3$O-P(4)$_{Mistral}$ | 81.4 | 0.55 | 3.95 | 82.6 | 0.56 | 5.24 | 96.2 | 0.52 | 1.52 | 66.6 | 0.53 | 2.41 | 57.4 | 0.52 | 3.04 |
| GeLLM$^3$O-P(4)$_{Llama}$ | 80.4 | 0.54 | 3.6 | 81.4 | 0.56 | 4.81 | 93.8 | 0.47 | 1.64 | 61.4 | 0.5 | 2.02 | 49.8 | 0.48 | 3.26 |
| GeLLM$^3$O-P(6)$_{Mistral}$ | 83.0 | 0.57 | 3.6 | 85.8 | 0.59 | 4.78 | 96.8 | 0.53 | 1.48 | 60.8 | 0.54 | 2.16 | 54.0 | 0.54 | 3.09 |
| GeLLM$^3$O-P(6)$_{Llama}$ | 77.0 | 0.53 | 3.73 | 79.6 | 0.56 | 5.05 | 95.0 | 0.47 | 1.66 | 57.0 | 0.49 | 2.50 | 52.2 | 0.49 | 3.48 |
| *Liquid Foundation Models* | | | | | | | | | | | | | | | |
| LFM2-2.6B | 38.0 | 0.18 | 1.16 | 31.4 | 0.08 | 0.88 | 80.2 | 0.19 | **2.09** | 32 | 0.11 | 0.76 | 17.4 | 0.11 | 1.62 |
| LFM2-2.6B-MMAI, MT | 86.0 | 0.42 | 4.09 | **91.6** | 0.50 | 4.54 | **98.8** | 0.41 | 1.82 | **71.2** | 0.40 | 2.72 | 56.4 | 0.38 | 3.67 |
| LFM2-2.6B-MMAI, ST | 86.0 | 0.41 | 4.35 | 91.2 | 0.51 | 4.42 | **98.8** | 0.41 | 1.88 | 69.8 | 0.39 | 2.95 | **57.8** | 0.37 | 4.00 |

Table 1: **Molecular Optimization results on MuMO-Instruct.** We report Success Rate (SR), Similarity (Sim), and Relative Improvement (RI) on five tasks. Best is **bold**, second best is underlined.

When evaluating base models that have not been fine-tuned on our instruction format, we utilize prompts provided by authors or prepend few-shot examples randomly sampled from the training split of the same task to guide the model on the expected input and output formatting.

In all benchmarks, we report SFT+Multi-task RFT (**MT**) and SFT+Single-Task RFT (**ST**) models with enabled (`/think`) and disabled(`/no_think`) reasoning on inference across ADME/PK and toxicity tasks. The only exception is MuMO-Instruct, where we report the non-reasoning setting only to ensure a fair comparison with prior baselines.

## 4.2 MuMO-Instruct Results

**Task, Setup, and Metrics.** We evaluate our model's capability in multi-objective lead optimization using the MuMO-Instruct benchmark (Liu et al., 2025). Unlike standard single-property tasks, MuMO-Instruct requires the model to modify a starting molecule to simultaneously improve multiple conflicting properties while retaining structural similarity. We report results on five core molecular property combinations: BDP (BBBP, DRD2, PlogP), BDQ (BBBP, DRD2, QED), BPQ (BBBP, PlogP, QED), DPQ (DRD2, PlogP, QED), and the four-property task BDPQ. Performance is measured using three metrics: Success Rate (SR), the percentage of generated molecules that satisfy all property optimization constraints; Similarity (Sim), the Tanimoto similarity between the input and generated optimized molecule; and Relative Improvement (RI), quantifying the magnitude of property enhancement.

**Baselines.** We benchmark LFM2-2.6B-MMAI against a comprehensive suite of models: Foundational Chemistry LLMs; Task-specific non-LLMs and LLMs; Generalist LLMs, which represent the current state-of-the-art for this benchmark.

**Summary of Results.** Table 1 shows that LFM2-2.6B-MMAI substantially improves over the base LFM2-2.6B across all task settings. Across the three-property tasks, LFM2-2.6B-MMAI achieves best (or near-best) Success Rate, matching or surpassing specialized GeLLMO variants, and it remains strong on the more challenging four-property BDPQ setting where it achieves the best overall Success Rate among all compared methods. While generalist baselines can preserve higher structural similarity, their substantially lower Success Rates indicate limited ability to satisfy the joint pharmacological constraints. In contrast, LFM2-2.6B-MMAI strikes a more effective balance between making meaningful structural edits (moderate similarity) and consistently meeting the multi-property optimization requirements, while also delivering strong RI on several tasks.

| Model | Single | | | | Interaction | | | | Comparison | | | |
|---|---|---|---|---|---|---|---|---|---|---|---|---|
| | Boolean | | Value | | Boolean | | Value | | Boolean | | Value | |
| | Acc | Val | RMSE | Val | Acc | Val | RMSE | Val | Acc | Val | RMSE | Val |
| *Proprietary LLMs* | | | | | | | | | | | | |
| GPT-5-1 | 0.687 | **1.000** | 88.055 | **1.000** | 0.526 | **1.000** | 41.062 | **1.000** | 0.693 | **1.000** | 63.233 | **1.000** |
| GPT-5-2 | 0.741 | **1.000** | 88.942 | **1.000** | 0.531 | **1.000** | 42.624 | **1.000** | 0.637 | **1.000** | 66.223 | **1.000** |
| Sonnet-4-5 | 0.663 | **1.000** | 92.274 | **1.000** | 0.606 | 0.999 | 45.147 | **1.000** | 0.741 | **1.000** | 64.468 | **1.000** |
| Opus-4-5 | 0.761 | **1.000** | 105.049 | **1.000** | 0.729 | **1.000** | 51.932 | **1.000** | 0.777 | **1.000** | 71.679 | **1.000** |
| *Open-weight LLMs* | | | | | | | | | | | | |
| DeepSeek-3-2 | 0.629 | 0.963 | 68.866 | 0.424 | 0.476 | 0.896 | 44.610 | 0.260 | 0.494 | 0.994 | 55.504 | 0.643 |
| Llama-3.1 8B | 0.548 | 0.993 | 162.351 | 0.840 | 0.547 | 0.982 | 421.325 | 0.780 | 0.474 | 0.991 | 80.566 | 0.761 |
| Llama-3.1 70B | 0.683 | **1.000** | 84.119 | 0.973 | 0.530 | **1.000** | 38.646 | 0.977 | 0.456 | **1.000** | 64.887 | 0.943 |
| Qwen2.5-7B | 0.590 | 0.999 | 63.511 | 0.576 | 0.396 | 0.999 | 36.307 | 0.683 | 0.664 | **1.000** | 65.471 | 0.223 |
| *Foundational LLMs for Chemistry* | | | | | | | | | | | | |
| ChemLLM-7B | 0.233 | 0.997 | 209.584 | 0.629 | 0.235 | 0.997 | 162.742 | 0.566 | 0.250 | **1.000** | 65.428 | 0.514 |
| Llama-3-8B-MolInst | 0.107 | 0.203 | 328.935 | 0.496 | 0.059 | 0.149 | 188.376 | 0.486 | 0.469 | 0.873 | 138.654 | 0.837 |
| LlaSMol-Mistral-7B | 0.387 | 0.922 | 266.720 | 0.923 | 0.298 | 0.968 | 262.550 | 0.983 | 0.239 | **1.000** | 245.298 | 0.924 |
| *Liquid Foundation Models* | | | | | | | | | | | | |
| LFM2-2.6B | 0.740 | **1.000** | 69.368 | **1.000** | 0.724 | **1.000** | 38.905 | **1.000** | 0.712 | **1.000** | 61.556 | **1.000** |
| LFM2-2.6B-MMAI, MT, `/no_think` | 0.838 | **1.000** | 62.766 | **1.000** | 0.808 | **1.000** | 31.018 | **1.000** | 0.789 | **1.000** | 53.127 | **1.000** |
| LFM2-2.6B-MMAI, MT, `/think` | 0.840 | **1.000** | 58.880 | **1.000** | 0.815 | **1.000** | 27.135 | **1.000** | 0.801 | **1.000** | **48.344** | **1.000** |
| LFM2-2.6B-MMAI, ST, `/no_think` | **0.841** | **1.000** | 62.902 | **1.000** | 0.808 | **1.000** | 30.248 | **1.000** | 0.796 | **1.000** | 53.045 | **1.000** |
| LFM2-2.6B-MMAI, ST, `/think` | 0.840 | **1.000** | **55.954** | **1.000** | **0.819** | **1.000** | **25.046** | **1.000** | **0.810** | **1.000** | 50.598 | **1.000** |

Table 2: **Functional Group Reasoning results on FGBench.** We report accuracy **Acc** for questions, **RMSE** for regression, and fraction of valid predictions (**Val**). Best is **bold**, second best is underlined.

## 4.3 FGBench Benchmark Results

**Task, Setup, and Metrics**. The FGBench Liu et al. (2025) is an instruction-following benchmark for functional-group editing, a key task that reflects molecular property reasoning. Each example in the dataset consists of a reference molecule and a textual description of its modification (e.g., adding or removing named functional groups at specified positions) and asks the model to predict how a specified property changes under that edit. The dataset supports three types of questions: the effect of modifying a single functional group, the effect of modifying multiple groups, and the question of predicting a specified property for the target molecule given a similar reference molecule and its value of the property. Each group of questions comes in True / False and numeric value prediction variants. We calculate accuracy (**Acc**) for the binary tasks and **RMSE** for regression.

**Baselines.** We compare the performance of two LFM2-2.6B-MMAI models against several baseline models: a) the original LFM-2 model prior to MMAI gym; b) proprietary general-purpose LLMs; c) open-weight general-purpose LLMs; and d) chemical specialist LLMs.

**Summary of results.** As shown in Table 2, LFM2-2.6B-MMAI improves substantially over the base LFM2-2.6B across all three FGBench regimes (Single, Interaction, and Comparison) and for both boolean classification and numeric value prediction, demonstrating that MMAI Gym training strengthens functional group reasoning. Across our variants, /think tends to be most beneficial for the numeric (RMSE) setting—especially on the more challenging Interaction and Comparison questions—while /no_think remains a strong, lower-compute option for the boolean tasks. Overall, LFM2-2.6B-MMAI is competitive with the strongest proprietary general-purpose LLMs on boolean accuracy and achieves state-of-the-art performance on the value-prediction portions of FGBench, while chemical specialist LLMs lag behind across the suite.

## 4.4 Single-Step Retrosynthesis Results

**Task, Setup, and Metrics.** We evaluate our models for the ability to perform a single-step retrosynthesis (SSRS) task (Choe et al., 2025), a standard task related to full-scale synthesis planning; on the conventional and curated USPTO-50K test benchmark (Liu et al., 2017) based on public data; and the URSA-expert-2026 benchmark (Zagribelnyy et al., 2026), which contains out-of-distribution (OOD) target molecules with undisclosed answers. ChemCensor (CC) Zagribelnyy et al. (2026), a recently introduced data-driven metric for evaluating chemical plausibility based on synthetic precedents, were used to benchmark the models' performance in the SSRS task.

**Baselines.** We compare the performance of the LFM2-2.6B-MMAI models against several baseline tiers: a) proprietary general-purpose (GP) LLMs; b) open-weight GP LLMs; c) chemical generalist LLMs, and d) the original LFM2-2.6 model prior to the domain-specific training.

| Model | URSA-expert-2026 | | | | | USPTO-50K-test | | | | |
|---|---|---|---|---|---|---|---|---|---|---|
| | Unique | Max | Av. PT-Top-K CC @3 | @5 | @10 | Unique | Max | Av. PT-Top-K CC @3 | @5 | @10 |
| *Proprietary LLMs* | | | | | | | | | | |
| Grok-4.1 | 43% | 1.75 | **1.29** | **0.93** | **0.48** | – | – | – | – | – |
| Gemini 2.5 Flash | 50% | 0.66 | 0.30 | 0.18 | 0.09 | – | – | – | – | – |
| Gemini 3 Flash preview | 40% | **1.80** | 1.21 | 0.83 | 0.42 | – | – | – | – | – |
| GPT 5.1 | 62% | 0.73 | 0.35 | 0.22 | 0.11 | 59% | 1.54 | 0.72 | 0.45 | 0.22 |
| GPT 5.2 | 51% | 0.85 | 0.47 | 0.29 | 0.15 | 52% | 2.04 | 1.01 | 0.63 | 0.32 |
| Claude 4.5 Sonnet | 70% | 1.45 | 0.88 | 0.58 | 0.29 | 56% | 3.37 | 1.81 | 1.15 | 0.58 |
| Claude 4.5 Opus | 54% | 1.34 | 0.73 | 0.46 | 0.23 | 44% | 3.31 | 1.62 | 1.00 | 0.50 |
| Claude 4.6 Opus | 30% | 1.39 | 0.87 | 0.56 | 0.28 | – | – | – | – | – |
| *Open-weight LLMs* | | | | | | | | | | |
| DeepSeek 3.2 | 52% | 0.39 | 0.16 | 0.10 | 0.05 | 55% | 1.14 | 0.46 | 0.27 | 0.14 |
| Qwen3 8B | 52% | 0.00 | 0.00 | 0.00 | 0.00 | 55% | 0.04 | 0.03 | 0.02 | 0.01 |
| Qwen3 14B | 52% | 0.01 | 0.00 | 0.00 | 0.00 | 54% | 0.10 | 0.03 | 0.02 | 0.01 |
| Kimi K2 | 43% | 1.12 | 0.62 | 0.38 | 0.19 | – | – | – | – | – |
| Kimi K2.5 | 50% | 1.49 | 1.01 | 0.66 | 0.34 | – | – | – | – | – |
| *Open-weight Chemical Specialist Models* | | | | | | | | | | |
| ether0 | 20% | 1.01 | 0.55 | 0.35 | 0.17 | 19% | 2.08 | 1.07 | 0.67 | 0.34 |
| NatureLM | 27% | 1.57 | 0.97 | 0.61 | 0.30 | 20% | 3.99 | 1.85 | 1.14 | 0.57 |
| RetroDFM-R | 12% | 1.52 | 0.70 | 0.43 | 0.21 | 8% | **4.35** | 1.56 | 0.94 | 0.47 |
| *Liquid Foundation Models* | | | | | | | | | | |
| LFM2-2.6B | 0% | 0.00 | 0.00 | 0.00 | 0.00 | 0% | 0.00 | 0.00 | 0.00 | 0.00 |
| LFM2-2.6B-MMAI Gym, MT, `/no_think` | **94%** | 1.18 | 0.67 | 0.42 | 0.21 | **90%** | 2.90 | 1.63 | 1.06 | 0.54 |
| LFM2-2.6B-MMAI Gym, MT, `/think` | **94%** | 1.51 | 0.94 | 0.61 | 0.31 | 89% | 3.16 | 1.87 | 1.26 | 0.64 |
| LFM2-2.6B-MMAI Gym, ST, `/no_think` | 78% | 0.99 | 0.58 | 0.37 | 0.18 | 72% | 2.46 | 1.44 | 0.95 | 0.48 |
| LFM2-2.6B-MMAI Gym, ST, `/think` | 93% | 1.51 | 1.00 | 0.68 | 0.34 | 88% | 3.27 | **1.96** | **1.32** | **0.67** |

Table 3: **Single-Step Retrosynthesis benchmarking results.** Columns report the following metrics. **Unique**: fraction of unique valid reactant sets among samples. **Max**: per-target maximum ChemCensor score averaged over targets. **Av. PT-Top-K CC**: per-target average ChemCensor score over top-K unique predictions. We highlight best values with **bold** and second best with underline.

**Summary of results.** As presented in the Table 3, the performance of purpose-trained LFM2-2.6B-MMAI models in solving the SSRS task boosted from 0 level to the levels comparable with top-tier proprietary GP and chemical generalist LLMs. The model with reasoning enabled consistently improves CC metrics over non reasoning model, indicating that chemist-like reasoning injection is beneficial for selecting more chemically credible disconnections. Among the LFM2 variants, the models with reasoning achieve the best overall performance, and the strongest setting attains state-of-the-art results on USPTO-50K-test on the CC-based aggregated metrics while remaining competitive on URSA-expert-2026. Overall, LFM2-2.6B-MMAI models outperform open-weight general-purpose and chemical specialist baselines on diversity and CC metrics, and are competitive with top-tier proprietary LLMs.

## 4.5 TDC Benchmark Results

**Task, Setup, and Metrics.** We evaluate models on the Therapeutics Data Commons (TDC) benchmark (Huang et al., 2021), a standardized platform for drug discovery tasks. We specifically focus on the absorption, distribution, metabolism, excretion (ADME), pharmacokinetics (PK), and toxicity groups, which require predicting pharmacological properties such as absorption (Caco2 cells monolayer permeability, human intestine absorption (HIA)), distribution (Blood-brain barrier (BBB) penetration, plasma protein binding in rat plasma), metabolism (cytochrome P450 enzymes), and safety (hERG inhibition, Ames genetoxicity test). Following the standard TDC protocol, we utilize task-specific metrics including Mean Absolute Error (MAE) for regression tasks, Spearman's Correlation for pharmacokinetics, and AUROC or AUPRC for classification tasks. The baselines are the established TDC State-of-the-Art (SOTA) Leaderboard (Huang et al., 2021) and a much larger general-purpose scientific model trained on TDC, TxGemma-27B (Wang et al., 2025).

**Baselines.** We compare the performance of LFM2-2.6B-MMAI models against several baseline tiers: a) LFM2-2.6B: the original autoregressive hybrid model without domain-specific training to isolate the impact of our MMAI Gym; b) TxGemma-27B (Wang et al., 2025): a heavyweight 27-

| Group | Task | Metric | Liquid Foundation Models | | | | | TxGemma-27B | TDC SOTA |
|---|---|---|---|---|---|---|---|---|---|
| | | | Base | MT | | ST | | | |
| | | | | /no_think | /think | /no_think | /think | | |
| ADME/PK | Caco2 Wang | MAE (↓) | 0.550 | 0.365 | 0.396 | 0.356 | 0.347 | 0.401 | **0.256** |
| | Lipophilicity AZ | MAE (↓) | 0.953 | 0.454 | 0.441 | 0.457 | **0.434** | 0.538 | 0.456 |
| | Solubility AqSolDB | MAE (↓) | 2.066 | 0.833 | 0.812 | 0.836 | 0.812 | 0.907 | **0.741** |
| | PPBR AZ | MAE (↓) | 10.130 | 7.722 | 7.755 | 7.917 | 8.006 | 9.048 | **7.440** |
| | VDss Lombardo | Spearman (↑) | 0.131 | 0.585 | 0.596 | 0.589 | 0.591 | 0.559 | **0.713** |
| | Half Life Obach | Spearman (↑) | -0.023 | 0.418 | 0.380 | 0.402 | 0.381 | 0.458 | **0.576** |
| | Clearance Microsome AZ | Spearman (↑) | -0.020 | 0.542 | 0.554 | 0.566 | 0.531 | 0.462 | **0.630** |
| | Clearance Hepatocyte AZ | Spearman (↑) | -0.036 | 0.372 | 0.377 | 0.370 | 0.356 | 0.260 | **0.536** |
| | Pgp Broccatelli | AUROC (↑) | 0.585 | 0.887 | 0.852 | 0.896 | 0.863 | 0.937 | **0.938** |
| | HIA Hou | AUROC (↑) | 0.428 | 0.981 | 0.987 | 0.977 | 0.987 | 0.988 | **0.993** |
| | BBB Martins | AUROC (↑) | 0.366 | **0.930** | 0.922 | **0.930** | 0.924 | 0.908 | 0.924 |
| | Bioavailability Ma | AUROC (↑) | 0.516 | 0.752 | 0.752 | 0.746 | 0.751 | 0.694 | **0.942** |
| | CYP2C9 Substrate CM | AUPRC (↑) | 0.346 | 0.427 | 0.427 | 0.426 | 0.444 | 0.438 | **0.474** |
| | CYP2C9 Veith | AUPRC (↑) | 0.372 | 0.774 | 0.752 | 0.772 | 0.751 | 0.683 | **0.859** |
| | CYP2D6 Substrate CM | AUPRC (↑) | 0.366 | 0.649 | 0.631 | 0.658 | 0.630 | 0.711 | **0.736** |
| | CYP2D6 Veith | AUPRC (↑) | 0.206 | 0.631 | 0.582 | 0.636 | 0.585 | 0.683 | **0.790** |
| | CYP3A4 Substrate CM | AUROC (↑) | 0.571 | **0.719** | 0.686 | 0.711 | 0.681 | 0.691 | 0.667 |
| | CYP3A4 Veith | AUPRC (↑) | 0.477 | 0.865 | 0.845 | 0.862 | 0.855 | 0.854 | **0.916** |
| Tox | LD50 Zhu | MAE (↓) | 0.799 | 0.706 | 0.709 | 0.711 | 0.703 | 0.627 | **0.552** |
| | DILI | AUROC (↑) | 0.612 | 0.950 | 0.941 | 0.947 | 0.937 | 0.886 | **0.956** |
| | hERG | AUROC (↑) | 0.679 | 0.779 | 0.789 | 0.784 | 0.787 | **0.885** | 0.880 |
| | AMES | AUROC (↑) | 0.522 | 0.805 | 0.792 | 0.800 | 0.782 | 0.816 | **0.871** |

Table 4: **ADMET Properties Prediction results on TDC.** We report the same base and LFM2-2.6B-MMAI model results after MMAI Gym. ST is single-category RFT, including all TDC ADMET tasks. TDC SOTA is the specialist models leaderboard. Best is **bold**, second best is underlined.

billion parameter language model tailored for therapeutics; c) TDC SOTA: the highest recorded performance on the official TDC leaderboard for each respective task.

**Summary of Results.** As shown in Table 4, LFM2-2.6B-MMAI consistently improves over the base LFM2-2.6B across essentially all ADME/PK and toxicity tasks, highlighting the benefit of MMAI Gym training. Across the Liquid model variants, the best-performing configurations are competitive with substantially larger models: they surpass TxGemma-27B on multiple ADME/PK task and are comparable on others. While we achieve performance on par with or superior to heavyweight LLMs, our model does not yet consistently beat the specialized TDC SOTA in regression tasks like Clearance or VDss, where specialist non-LLM models still maintain a lead. Across settings, multi-task RFT (MT) training is generally more robust and competitive across the full benchmark suite, while single-task RFT (ST) training can yield the strongest results on particular datasets. We further observe that enabling thinking often provides a small but consistent lift on harder endpoints, whereas no reasoning model is typically comparable on easier tasks and offers a lower-compute alternative with minimal loss in performance.

## 5 CONCLUSION

Our experiments show that specialist-level performance in drug discovery does not require frontier-scale model size. With the right combination of molecular data, training procedures, and efficient architectures, it is possible to train a competitive model across a wide range of practical tasks. For ADMET property prediction, we find that a smaller model trained in a multi-task setting across hundreds of tasks and dozens of categories can match or exceed models more than an order of magnitude larger, even when those larger baselines are fine-tuned on narrowly relevant, single-category tasks. In a few cases, the multi-task model also surpasses strong single-task specialist predictors, suggesting that breadth of training can improve accuracy and robustness rather than dilute performance.

We observe an even stronger pattern in molecular optimization and functional group reasoning, where the model achieves the highest success rates. These results indicate that the model learns actionable structure–property relationships that translate into effective molecular edits. Even in harder settings, such as single-step retrosynthesis, the smaller model remains competitive and produces results close to substantially larger models. When multi-task training alone does not yield state-of-the-art performance, results can be further improved through additional category-focused training, narrowing the remaining gap without sacrificing overall efficiency.

Taken together, these findings support the main message of this paper. MMAI Gym for Science provides a recipe that can substantially improve the capabilities of foundation models, delivering new state-of-the-art or near state-of-the-art results, as well as strong accuracy and efficiency trade-offs across core drug discovery tasks.

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

# A    TRAINING PROTOCOL

**SFT:**    At each optimization step of $100,000$, we process an aggregate context of $3,145,728$ tokens across all GPUs, using sequence packing Brown et al. (2020) to fill each device's available context window with multiple training examples. We conduct supervised fine-tuning on the MMAI Gym corpus with the `AdamW` optimizer, using a learning rate of $1\text{x}10^{-5}$, linear warmup for the first $5\%$ of the training steps, and a cosine decay schedule for the remainder.

**RFT:**    Online RFT was performed using Group Relative Policy Optimization (GRPO, Shao et al. (2024)), using the same tasks and prompts as in SFT. We use a group-size of $8$, with $16$ groups per global batch, resulting in a batch-size of $128$. With a maximum sequence length of $16,384$, this results in a maximum of $2,097,152$ tokens processed per step across all GPUs (we use one update per batch). However, the effective token throughput due to variable completion lengths was approximately $130,000$ tokens per step. Due to the significant number of SFT steps to fully learn molecular languages, we use a high sampling temperature of $1.4$ to encourage generation diversity. Also, due to the diversity of tasks and task-specific reward functions, we keep KL-regularization with $\beta = 0.4$. Concerning the optimizer, we use AdamW with a constant learning rate of $1\text{x}10^{-6}$ and weight-decay of $0.05$. Finally, we use Deepspeed stage 3 (Rasley et al., 2020), with gradient checkpointing and BF16 precision.

To explore the benefits of RFT, we either perform RFT on all tasks concurrently to train an RFT "generalist" model, or on a single task for a "specialist" model. For the generalist case, we sample prompts uniformly across all tasks, and use one task-specific reward function according to the sampled prompt (plus formatting rewards). Due to the high wall-time required to compute rewards for molecular generation tasks, all RFT setups involving generation were trained over $1,000$ steps.

