# OpenReview forum: "MMAI Gym for Science: Training Liquid Foundation Models for Drug Discovery"
_ICLR.cc/2026/Workshop/FM4Science — ICLR 2026 Workshop FM4Science Poster_

### Official Review · Reviewer_5dzJ · 2026-02-22
**MMAI Gym for Science: Training Liquid Foundation Models for Drug Discovery**

**Rating:** 8
**Confidence:** 4

**Review:**

### Summary
This paper introduces the MMAI Gym for Science, a comprehensive training and benchmarking environment specifically designed to teach foundation models the "language of molecules". The authors utilize this framework to fine-tune LFM2-2.6B, a highly efficient hybrid Liquid Foundation Model (LFM) that uses gated short convolution sequence operators to achieve sub-quadratic scaling and high throughput. The resulting model, LFM2-2.6B-MMAI, is evaluated across essential drug discovery tasks—including molecular optimization, ADMET property prediction, and retrosynthesis—where it demonstrates competitive or state-of-the-art performance, often surpassing substantially larger general-purpose models.

### Quality
The technical quality of the work is high, particularly in its architectural and training rigor. The authors leverage the LFM2-2.6B hybrid architecture, which achieves a favorable Pareto tradeoff between expressivity and efficiency by interleaving gated short convolutions with a small number of grouped-query softmax attention (GQA) layers. The training protocol is robust, utilizing a multi-stage curriculum of supervised fine-tuning (SFT) and reinforcement learning fine-tuning (RFT) using Group Relative Policy Optimization (GRPO). The inclusion of domain-specific reward functions, addressing formatting, reasoning length, and task-specific correctness, ensures the model learns "domain-faithful" reasoning chains.

### Clarity
The manuscript is well-organized and clearly articulates the limitations of general-purpose LLMs in scientific domains. The description of the MMAI Gym framework , spanning 200+ tasks across 2D/3D molecular and protein modalities, is detailed and easy to follow. However, some results tables, particularly those with numerous baselines, are quite dense and require careful inspection to interpret.

### Originality
The primary originality lies in the MMAI Gym framework itself, which moves beyond simple text prediction to treat drug discovery as a set of domain-specific reasoning tasks. The use of chemical format augmentation (e.g., SMILES to SELFIES to IUPAC) during training is a clever strategy to prevent overfitting to specific syntax and encourage the learning of abstract molecular concepts. While Liquid Foundation Models exist, applying this specific hybrid-LFM architecture to a multi-modal drug discovery reasoning suite represents a novel integration of efficient AI and medicinal chemistry.

### Significance
This work is significant for the field of computational drug discovery. It demonstrates that smaller, purpose-trained models (2.6B parameters) can match or exceed the performance of models an order of magnitude larger. The model's competitive performance on USPTO-50K-test for retrosynthesis and various TDC benchmarks suggests it can handle practical R&D workloads.

### Pros
Exceptional Efficiency: Achieves higher throughput than size-matched baselines and remains efficient even with long-context inputs.

Breadth of Training: Covers a massive scope of over 200 tasks across diverse molecular and protein modalities.

### Cons
Specialist Gap: Despite strong performance, the model does not yet consistently beat specialized non-LLM models on certain regression tasks like Clearance or VDss.

Parameter Scaling Limits: The study focuses on a 2.6B parameter model; it is unclear if the specific benefits of the LFM architecture continue to scale linearly for much larger scientific models.

---

### Official Review · Reviewer_gRkx · 2026-02-24
**Strong applied results but reads more as product demonstration than scientific contribution**

**Rating:** 5
**Confidence:** 4

**Review:**

This paper presents MMAI Gym, a training suite for adapting LLMs to drug discovery, and uses it to finetune the 2.6B parameter LFM2 model via SFT plus GRPO based reinforcement learning. Results across MuMO Instruct, FGBench, USPTO 50K retrosynthesis, and TDC ADMET benchmarks show the finetuned model matching or beating models 10x its size including TxGemma 27B and several proprietary frontier LLMs. The breadth of evaluation is commendable and the chemical format augmentation strategy converting between SMILES SELFIES and IUPAC during training is a nice practical contribution. The reward function design for classification, regression and generation tasks is also thoughtful. However the paper reads more as a product demonstration for Liquid AI and Insilico Medicine than a scientific contribution. The LFM2 architecture is taken from a prior report, the training recipe is standard SFT plus GRPO, and the key datasets include large amounts of proprietary in house data that cannot be inspected or reproduced. No ablations disentangle what drives the gains whether its data quality, volume, multi task balancing, tokenization or RFT. Anonymization is also weak since LFM2 and Chemistry42 references make author identity fairly transparent. The number of evaluation repetitions and variance across them are not reported and some state of the art claims rely on ChemCensor metrics introduced by what appears to be the same group.

---

### Official Review · Reviewer_m3hj · 2026-02-24
**Promising Benchmarking Framework for Domain Adaptation in Drug Design, Requiring Greater Clarity and Structural Refinement**

**Rating:** 4
**Confidence:** 5

**Review:**

The manuscript addresses an important problem: the limited generalization of general-purpose foundation models to specialized drug discovery tasks. To tackle this, the authors propose MMAI Gym for Science, a benchmarking framework that combines (i) an evaluation pipeline, (ii) reinforcement-based adaptation schemes for task specialization, and (iii) a collection of benchmark datasets. They further introduce an evaluation strategy intended to mitigate bias arising from heterogeneous data sources.

The work is information-rich and has clear potential. However, several aspects lack clarity, and the presentation appears tailored primarily to a chemically specialized audience, limiting accessibility for the broader molecular research community.

Comments

1.	Figure 2 is very dense and lacks a sufficiently explanatory caption. It is referenced only once in the text and not described in detail. While it appears to illustrate the overall framework, it is difficult to interpret without proper guidance.

2.	The first paragraph of Section 3 reads largely as introductory material and overlaps with content already presented in Section 1.

3.	In Section 3.1, numerous datasets are mentioned, but not rigorously specified (“including but not limited to”). It is unclear how these datasets relate to the six benchmarks described later.

4.	Section 3.1 refers to “six top-level categories,” yet their definition remains unclear. It is not specified whether a category corresponds to a benchmark suite or something else. Although Figure 2 may hint at this structure, it is not explicitly explained in the text.

5.	Section 3.2 is difficult to follow. Extensive data augmentation is described, presumably to regularize training, but the conceptual differences between SMILES, SELFIES, and IUPAC representations are not discussed. The focus remains on representational diversity rather than conceptual understanding.

6.	The benchmarking protocol states that each test example was evaluated multiple times, but the exact number of evaluations is not reported. It is also unclear whether the tables report means, medians, or another statistic. If means are reported, including variances would provide insight into performance stability.

7.	Different baseline sets are used across benchmarks. While some task-specific variation is understandable, certain general models (e.g., GPT-5–like models) could plausibly apply across multiple tasks. The rationale for their selective inclusion should be clarified.

8.	Evaluation metrics are only sparsely explained. While common metrics such as MAE or AUPR need no definition, others (e.g., Tanimoto similarity) would benefit from brief clarification. Additionally, e.g. the term “valid prediction” appears only in the caption of Table 2 without explanation of what constitutes validity.

9.	No system and GPU specification is provided (“across all GPUs”, although it is not even clear how many there were)

Overall, the manuscript presents promising ideas and results, but would benefit from substantial restructuring. Reducing redundancy, clarifying technical terminology for a broader audience, and improving logical flow would significantly strengthen the work.

---

### Official Review · Reviewer_gEcT · 2026-02-25
**This paper introduces MMAI Gym, a domain-adaptation framework that equips a compact 2.6B hybrid language model with chemistry-aware tokenization, structured augmentation, and supervised plus reinforcement fine-tuning to create a strong generalist for drug discovery tasks. The model achieves competitive performance across molecular optimisation, functional-group reasoning, retrosynthesis, and ADMET prediction, often matching those of larger, specialised models while maintaining efficiency. The work is technically sound, clearly presented, and practically significant, particularly in demonstrating that careful domain-aligned training can make small models highly competitive. However, concerns remain regarding reproducibility, transparency of contamination, statistical reporting, computational disclosure, and confidence in reward design choices. Overall, the paper makes a meaningful applied contribution with strong practical relevance but would benefit from greater rigor and transparency to fully substantiate its claims.**

**Rating:** 6
**Confidence:** 4

**Review:**

Overall Evaluation: This work presents MMAI Gym for Science, a domain-adaptation framework that transforms a compact 2.6B hybrid language model (LFM2) into a strong generalist for drug discovery through chemistry-aware tokenization, structured augmentation, supervised fine-tuning (SFT), and reinforcement fine-tuning (RFT) with task-specific rewards. The resulting model demonstrates competitive performance across molecular optimization, functional-group reasoning, retrosynthesis, and ADMET/PK/toxicity prediction, often matching or surpassing significantly larger models while maintaining strong inference efficiency.

Quality: The work is technically sound and methodologically coherent. The integration of format-aware tokenization (SMILES/SELFIES/FASTA), structure-preserving augmentation, and GRPO-based reinforcement learning is well-motivated and competently implemented. The experimental coverage is broad, including multiple task families and baselines, which strengthens the empirical credibility.

However, several issues limit complete confidence in the claims:
1. Lack of statistical reporting (variance across seeds, confidence intervals).
2. Limited quantitative contamination analysis despite widespread use of large-scale public corpora.
3. Partial reliance on proprietary benchmarks reduces reproducibility.
4. Missing compute budget disclosures.

Overall quality is high, but could benefit from stronger reproducibility and greater statistical rigour.

Clarity: The paper is clearly written and logically structured. The training pipeline, reward design, and architectural decisions are explained with reasonable detail. Benchmarking protocols are sufficiently described for a high-level understanding. Minor formatting inconsistencies and insufficient compute/resource reporting slightly reduce polish, but clarity is generally intense.

Originality: The contribution is not centered on a fundamentally new architecture, but rather on a carefully engineered, domain-faithful training suite that combines multiple pragmatic innovations:

1. Chemistry-specific tokenization and traversal augmentation.
2. Cross-format molecular representation handling.
3. Reinforcement fine-tuning with domain-aware reasoning signals.
4. Emphasis on efficiency via a hybrid ShortConv + sparse GQA backbone.

While individual components are not entirely novel, their integration into a unified, generalist molecular reasoning framework represents meaningful applied originality.

Significance: The central claim, that a compact, efficiency-oriented model can achieve near-specialist performance across diverse drug discovery tasks, is practically essential. If validated with more substantial evidence of reproducibility and released publicly, MMAI Gym could become a valuable community benchmark and training pipeline.

The efficiency-performance trade-off is especially significant in academic labs and real-world deployment scenarios with limited compute resources.

Pros:
1. Broad multi-task evaluation across optimization, reasoning, retrosynthesis, and ADMET.
2. Demonstrates strong efficiency–performance trade-offs.
3. Well-justified chemistry-aware tokenization and augmentation strategy.
4. Reinforcement fine-tuning yields consistent improvements over SFT-only.
5. Includes OOD considerations and template-level evaluation.
6. Practical relevance for real-world drug discovery workflows.
7. Clear and mostly well-structured presentation.

Cons:
1. No statistical significance analysis or multi-seed reporting.
2. Limited transparency on contamination audits and overlap statistics.
3. Partial reliance on proprietary or non-reproducible benchmarks.
4. Training compute and resource budgets are not disclosed.
5. RL reward design may incentivise verbosity (a length-based “thinking” bonus).
6. Fragile probability extraction via first-token logprobs.
7. Comparisons with classical ADMET baselines and recent optimization frameworks are incomplete.
8. Ethical and dual-use considerations are under-discussed.

Final Assessment: This is a strong applied systems paper demonstrating that carefully designed domain adaptation and reinforcement fine-tuning can make compact language models competitive with much larger systems in molecular reasoning tasks. The work is technically credible, practically significant, and clearly presented.
However, improvements in reproducibility, transparency about contamination, statistical rigour, and computational disclosure are necessary to substantiate the strength of its claims fully.

---

### Official Review · Reviewer_UED2 · 2026-02-25
**review for MMAI Gym for Science: Training Liquid Foundation Models for Drug Discovery**

**Rating:** 7
**Confidence:** 3

**Review:**

This paper introduces MMAI-Gym, an environment for training and evaluating multi-modal AI agents in scientific workflows. The framework integrates simulation tools, structured task definitions, and evaluation metrics to benchmark agent reasoning, planning, and tool use in science-oriented settings. The authors demonstrate the platform across multiple scientific domains and report baseline results using LLM-based agents.


Strengths
	•	Timely problem: Systematic benchmarking of AI agents for scientific reasoning and tool interaction is important and underdeveloped.
	•	Unified infrastructure: The environment integrates simulation, task orchestration, and evaluation in a coherent framework.
	•	Multi-modal focus: Support for structured data, text, and tool interaction reflects realistic scientific workflows.

Weaknesses
	•	Limited methodological novelty: The main contribution is infrastructure rather than new algorithms.
	•	Evaluation depth: Baseline experiments are relatively shallow; stronger agent comparisons (e.g., advanced tool-augmented models) would strengthen claims.
	•	Task diversity clarity: More quantitative characterization of task difficulty and distribution would help assess benchmark coverage.
	•	Scalability discussion is limited (e.g., computational cost, extensibility to new domains).

Overall Evaluation

A useful and well-motivated infrastructure paper addressing an important gap in scientific AI benchmarking. While algorithmic novelty is limited and experimental depth could be stronger, the platform has clear practical value and could serve as a foundation for future research.

---

### Official Review · Reviewer_wctg · 2026-02-26
**MMAI Gym for Science: Great multi-task training for molecular LLMs with some transparency and reproducibility gap**

**Rating:** 7
**Confidence:** 4

**Review:**

The paper introduces MMAI Gym for Science, an environment for training and benchmarking molecular foundation models. It uses a 2.6B parameter Liquid Foundation Model (LFM2), and applies multi-modal data augmentation, supervised finetuning (SFT), and reinforcement learning finetuning (RFT).

The authors report results on molecular optimization, retrosynthesis, and more and show that their model can compete or outperform other/larger models like TxGemma-27B. But the fact that it's dependent on proprietary data brings up questions about scientific transferability. Additionally, the ensemble of complex evaluation strategies makes it hard to pinpoint the exact source of performance gains by the model.

Overall though, the work is of high quality. The clarity of the training curriculum, moving from SFT to task-specific and multi-task RFT is interesting and admirable. The originality of this work is specially in the MMAI Gym curriculum, which frames chemical tasks as reasoning chains, e.g., using <think> tags, rather than simple text prediction.

Pros:
- It's efficient. the 2.6B model in this work outperforms models that are 10 times bigger in size (e.g., TxGemma-27B) as a result of better data processing.
- Using smart daata augmentations (random SMILES traversal) to avoid overfitting
- Good evaluation metrics: using ChemCensor for retrosynthesis instead of exact/hard ground-truth
- Generalization of the method: it performs well on out-of-distribution benchmarks (URSA=expert-2026)

Cons:
- The work is dependent on large proprietary data which makes it hard for the research community to independently verify the results.
- The model fails to beat non-LLM specialist models in regression tasks (Clearance, VDss)

Overall, it is a strong paper for the workshop. It shows the power of domain-specific RFT.

---

### Decision · Program_Chairs · 2026-03-03

Accept (Poster)